



# Low salinity as a biosecurity tool for minimizing biofouling in ships sea-chests

Castro, Maria Cecilia T.[a,b,c] Vance, Thomas[d]; Yunnie, Anna L.E.[d]; Fileman, Timothy W.[d]; Hall-Spencer, Jason M.[b,e]

a Plymouth Marine Laboratory, Prospect Place, Plymouth, PL1 3DH, United Kingdom.

b School of Biological and Marine Sciences, University of Plymouth, Drake Circus, Plymouth, PL4 8AA, United Kingdom.

c Directorate of Ports and Coasts, Navy of Brazil, Rua Teófilo Otoni, 4,CEP 20090-070, Rio de Janeiro / RJ, Brazil.

d PML Applications Ltd, Prospect Place, Plymouth, PL1 3DH , United Kingdom.

e Shimoda Marine Research Centre, University of Tsukuba, 5-10-1 Shimoda City, Shizuoka 415-0025, Japan.

*Correspondence to :* Maria Cecilia T de Castro (mctcastro@yahoo.com)

**Abstract.** Biofouling is a major vector in the transfer of non-native species around the world. Species can be transported on virtually all submerged areas on ships (e.g. hulls, sea-chests, propellers) and so antifouling systems are used to reduce fouling. However, with increased regulation of biocides used in antifoulants (e.g. the International Maritime Organization tributyltin ban in 2008), there is a need to find efficient and sustainable alternatives. Here, we tested the hypothesis that short doses of low salinity water could be used to kill fouling species in sea-chests. Settlement panels were suspended at 1.5 m depth in a Plymouth marina for 24 months by which time they had developed mature biofouling assemblages. We exposed these panels to three different salinities (7 psu, 20 psu and 33 psu) for two hours using a model sea chest placed in the marina and flushed with freshwater. Fouling organism diversity and abundance was assessed before panels were treated, immediately after treatment, and then one week and one month later. Some native ascidian *Dendrodoa grossularia* survived, but all other macrobenthos were killed by the 7 PSU treatment after one week. The 20 PSU treatment was not effective at killing the majority of fouling organisms. On the basis of these results we propose that sea-chests be flushed with freshwater for at least two hours before ships leave port. This would not cause unnecessary delays or costs and could be a major step forwards in improving biosecurity.

## 1 Introduction

Biofouling is a major vector in the transfer of non-native species around the world (Carlton et al., 1995; Ruiz et al., 1997; Gollasch, 2002; Coutts & Taylor, 2004; Castro et al., 2017). Species can be transported on virtually all submerged areas of ships so anti-fouling systems are used. However, some areas on ships hulls, such as sea chests and chain lockers, are difficult to access and coat with anti-foulants. Consequently, these areas often get heavily fouled by a wide variety of marine organisms such as hydroids, serpulid polychaetes, barnacles, mussels, bryozoans and tunicates (Coutts & Taylor, 2004; Murray et al., 2011).





Non-native species introduction and spread is increasing, e.g. due to the opening of new trade routes, climate change
and the increasing speed of vessels. The International Maritime Organization (IMO) decided to tackle this problem
initially by adopting a set of voluntary regulations. In 2011, the IMO Marine Environment Protection Committee
issued Resolution MEPC.207(62) outlining measures to minimize the risk associated with ship biofouling. These
regulations are directed at many stakeholders (e.g. States, shipmasters, operators and owners, shipbuilders, port
authorities, ship repair, dry-docking and recycling facilities, anti-fouling paint manufacturers / suppliers). Two
subsequent sets of guidance on biofouling have since been released: one for recreational craft less than 24 meters in
length (MEPC.1/Circ.792, 2012), and the second evaluating the 2011 Guidelines for the control and management of
ship biofouling to minimize the transfer of invasive aquatic species (MEPC.1/Circ.811, 2013) (Castro, 2014).
Following the entry into force of the Ballast Water Convention in 2017, it seems probable that ship biofouling may
soon become the subject of a new international treaty. In May, 2017, a programme called "Building Partnerships to
Assist Developing Countries to Minimize the Impacts from Aquatic Biofouling" (or "GloFouling Partnerships" )
was approved by the Global Environment Facility to be implemented by the United Nations Development
Programme and executed by the IMO. An implementation phase will start in the second half of 2018 and last five
years (IMO Circular Letter No 3768).  In some countries, biofouling management plans and record books are
already in place as part of national regulations (e.g. in the United States of America, Australia and New Zealand).
For instance, in the State of California (USA), ship owner/operators of vessels of 300 gross tons or larger need to
answer eleven questions about hull husbandry every year (Scianni et al., 2013).
Biofouling increases shipping operational costs; even microbial fouling, which is a pre-cursor to macro-fouling,
increases fuel consumption due to frictional drag.  There are also the costs of hull cleaning and painting (Schultz et
al., 2011; Dobretsov et al., 2013; Davidson et al., 2016). Some organisms (e.g. bryozoans) are tolerant to antifouling
compounds and can grow on freshly applied antifouling paint, and are subsequently used as a substratum for other
species (Murray et al., 2011). With the ban of tributyltin in 2008, other anti-fouling systems started to be used.
Antifouling compounds have been developed from marine bacteria, cyanobacteria, fungi as well as eukaryotic
organisms (Dobretsov et al., 2013). Glycerophospholipids from soybeans are also effective booster biocides in
antifouling paint (Batista et al, 2015). In terms of mechanical tools to remove biofouling, Hearin et al. (2016)
showed that mechanical grooming is helpful in reducing fouling on submerged surfaces coated with fouling-release
coatings.
Niche areas on vessel hulls (e.g. gratings and propellers) represent a great challenge to minimising biofouling. On
larger vessels, sea chests maximize seawater inflow (e.g. for internal cooling systems and ballast water). These box-
shaped structures are difficult to access and coat, they have edges and welds that provide sheltered areas for
organisms to settle and recruit (Coutts & Dodgshun 2007). In Canada, a study of 82 sea-chests from commercial
ships showed that 80% of them had fouling organisms and that almost half had non-native species (Frey et al.,
2014).

Setting biosecurity goals and implementing measures for controlling non-indigenous species helps to avoid their
spread (Collin et al., 2015). In order to control biofouling in niche areas on ships, a simple efficient treatment
method is needed.  Numerous methods are available, for example ultraviolet light (Titus & Ryskiewich, 1994),



heated water and steam, (Leach, 2011; Piola & Hopkins, 2012; Growcott *et al*., 2016) or soaking areas in acids (e.g.
acetic acid) or alkalines, such as hydrated lime (Rolheiser  *et al.* (2012). In Alaska, the invasive colonial ascidian
*Didemnum vexillum* was exposed to various treatments using acetic acid, bleach, freshwater or brine with 100%
mortality when exposed to freshwater for four hours (McCann *et al*., 2013).  In Brazil, Moreira *et al.* (2014) tested
the use of freshwater to combat the spread of invasive corals *Tubastraea tagusensis* and *T. coccinea*. For both these
species, two hour exposure to fresh water killed all the corals and this treatment is now routinely used for combat the
spread of *Tubastraea* spp. on oil industry infrastructure. In New Zealand, Jute and Dunphy (2016) showed that two
hour exposure to fresh water killed the invasive Mediterranean fan worm *Sabella spallanzanii*, while in hypersaline
conditions (50 psu) 100% mortality was reached after 24 hours. Finally two studies conducted in Plymouth, UK,
showed that low saline treatments can be highly effective at reducing biofouling and can be used in conjunction with
anti-fouling coating systems (Minto, 2014; Quinton, 2014). Although chemical treatments, the use of heat, or the use
of UV light all work they can be costly, or pose health and safety risks and also increase corrosion of hulls. On the
other hand, freshwater is not dangerous, and it is cheap and widely available.
Given the importance of biofouling as a vector in the world transfer and spread of non-native species, this study
tested the hypothesis that a low saline environment can kill fouling species and offers a simple and efficient
biosecurity management tool to minimize biofouling in ship sea-chests.

**2 Methods**

**2.1 Study area**
An experiment was conducted in two phases, the first in November 2016 and the second in July/August 2017 in
Millbay Marina (50º21'47''N; 004º09'02''W), Plymouth, UK. The marina is tidal and open to Plymouth Sound, a
large bay on the south coast of Devon (SW England) that is sheltered by an artificial breakwater (Bremekamp,

97  2012).


**2.2 Research design**
A model sea-chest was built to find out the lowest steady salinity that could be achieved when the chest was flushed
with freshwater whilst submerged and open to surrounding seawater. The sea-chest was a polypropylene 80 l
container (external dimensions: 600 x 400 x 420 mm); 12 panels were fixed inside with stainless threaded rods to
simulate gratings. A YSI 556 Multiparameter meter, complete with conductivity probe, was hooked inside the box to
measure salinity. The box was deployed so that the panels were vertical and about 1.5 m from the seawater surface;
measurements of temperature and salinity started immediately after the deployment and were recorded every 10
seconds. To create a hyposaline environment inside our immersed sea-chest, freshwater was flushed into the box
through a hose connected to a tap on the pontoon. A flow rate of approximately 8 l/min was kept during the
experiment bearing in mind the necessity of preventing excess turbulence inside the box. Flow was suspended after
86 minutes when the salinity stabilised and the probe stopped recording five hours later.



Polyvinyl chloride (PVC) settlement panels (each 12 x 12 x 0.5 cm) were deployed in the same marina > two years
before the experiment, in June, 2014. Initially they were fixed in grids horizontally orientated with the roughened
side facing outwards, in a depth of approximately 1.5 m, avoiding sedimentation and algae growth (Quinton, 2014).
Five months before the low salinity experiment, panels were rearranged in a vertical position tied to a rope and
attached to the pontoon. At this stage, panels were less exposed to the light, almost under the pontoon which also
helped to preclude macroalgae. Fifteen of these panels were selected based on the existence of a well-developed
fouling community, including the native ascidian *Dendrodoa grossularia* on all panels and the non-native encrusting
bryozoan *Watersipora subatra* on most of the panels. The objective was to examine the effects of low salinity water
treatments on the whole community assemblage on each panel.
Panels were subjected to one of the following treatments: 7 psu, 20 psu and control (33 psu) for two hours (five
panels per treatment). The lowest salinity (7 psu) was chosen as it was the lowest steady value achieved inside our
simulated sea-chest. The exposure time was chosen based on the studies conducted by Moreira *et al.* (2014) and Jute
& Dunphy (2016). On the day before the experiment started, water from the marina was collected and stored in a
constant temperature room at Plymouth Marine Laboratory. The water used to prepare the different salinity
treatments during the experiment was a mix of local sea water and pure fresh water (Milli-Q water), stored in the
same room.

**2.3 Analysis**
An acrylic 12 x 12 cm quadrat divided into a 1 cm$^2$ square grid was used to enumerate organisms on the settlement
panels. The apparatus (settlement panel & quadrat) were submerged in seawater in a Pyrex dish for analysis. At each
intersection point on the grid, organisms were identified, where possible to species level. Each taxon was
enumerated, with colonial invertebrates counted as one maximum per square. Analysis times were set to a maximum
of 25 minutes in order to minimise stress to the organisms. Panels were evaluated regarding the abundance and
mortality of fouling organisms before the exposure to fresh water, immediately afterwards, and on two more
occasions: one week and one month after. Mortality was assessed e.g. through detachment of the organisms from the
panels, a lack of response (e.g. tunicates with no reaction when siphons were touched), absence of zooids in erect
bryozoans, alterations in the texture / colour of the organisms.
Data from fouling communities were entered into PRIMER-E for abundance analysis and were square root
transformed prior to clustering analysis according to Clarke et al., 2016. Dendrogram plots were used to determine
similarity of fouling communities before, immediately after, one week and one month after the exposure to one of
three salinities targeted by this experiment.

**3 Results**
The first phase of the experiment was to ascertain the lowest salinity that could be maintained inside our simulated
sea-chest. The salinity was initially 32 psu, decreasing to 24 psu after 25 mins, to 9 psu after 60 mins before
stabilizing at 7 PSU from 86 mins onwards. Once the freshwater supply was switched off the salinity inside the sea-





chest increased slowly over a 5 h 20 min period to 27.3 PSU, when the recordings ended. During this time the water
temperature varied between 13 and 13.6ºC.
Biofouling communities were similar on panels before and immediately after treatment but thereafter there were
marked differences since low salinity treatments killed most of the organisms present. Cluster analysis of the
biofouling community composition one week after the treatment (Fig. 1, and one month after, not shown) showed
that panels submitted to the same treatment were clustered together, as they had similar communities present. Tight
clustering was found for panels exposed to 7 psu; few mortality effects were found at 20 psu and no effects were
found on control panels (33 psu).
Figure. 1. Dendrogram showing significant separation between biofouling communities grown on settlement panels
treated with 7 psu and all the others treated with 20 psu and 33 psu (n=5 for each treatment).

On panels treated with 7 psu terebellid worms quickly disintegrated and the erect native bryozoan *Bugula neritina*
leached a purple/brown colour into the water. The native ascidian *Ciona intestinalis* was less reactive when touched
with forceps than before the exposure. Neither *Dendrodoa grossularia*, the most frequent organisms on all panels,
nor *Watersipora subatra* colonies showed immediate visual responses to the treatments. After one week levels of
mortality were much more noticeable: for example 142 *D. grossularia* were counted on the five panels submitted to
7 psu - after a week 52 of these disintegrated when touched and were clearly dead. Erect bryozoans fell apart when
touched with forceps and all of the *Ciona intestinalis* had fallen off the panels. All of the native ascidian *Ascidiella*
*mentula*, were killed by the 7 psu treatment and had lost colour with flaccid tests filled with a dark liquid of rotting
tissue. Most organsims exposed to the 33 or 20 psu treatments survived (Fig. 2). More grid squares with bare panel
or biofilm were counted on all panels treated with 7 PSU (Table 1). All *W. subatra* individuals were dead after a
week with dark slime covering the panels and the distinct odour of rotting organisms.

Figure. 2. A) Settlement panel one week after exposure to a 33 psu treatment showing the high biomass and diverse
biofouling community that had developed over two years at 1.5 m depth in a marina off Plymouth, UK.  B) Example
of a panel one week after exposure to a 20 psu treatment with many members of the biofouling community still
alive.  C) Panel one week after a 7 psu treatment showing black sulphurous rotting tissues. D) Typical panel
appearance one month after exposure to 7 psu showing a much reduced fouling community.

In the 20 psu exposures *C. intestinalis* were less responsive immediately after treatment. After one week, 50% of *W.*
*subatra* colonies were dead, of a total of 60 *D. grossularia* only two (3.3%) had died. Many *D. grossularia*
individuals were covered with *Diplosoma listerianum,* not previously observed. This colonial tunicate is widespread
in the United Kingdom and shows rapid reproduction and growth rates (Bullard *et al*., 2004, 2007; Vance *et al*.,

2009).

One month after exposure to the three salinity treatments there were still very clear differences among the treatment
groups although some recolononisation had begun on the 7 psu panels (Table 1).  Numbers of species and Shannon-
Wiener diversity index show a decrease in diversity after one week and a small increase after one month for panels
exposed to 7 psu (Fig. 3).



Table 1: Average number of biofouling individuals per panel subjected to treatment with 7 psu, 20 psu and 33 psu (control) water, showing % change in abundance after one week and after one month.

Figure 3. A) Average number of species and B) Shannon-Wiener diversity index (H') of two year old biofouling communities developed on PVC panels at 1.5 m depth in a marina off Plymouth, UK. Panels exposed to 7, 20 and 33 psu (Control) before treatment (ST), immediately after exposure (AF), one week after (1W) and after one month of exposure (1M). Error bars are ± SD, n=15.

**4 Discussion**

We obtained a steady value of 7 psu inside our model sea-chest when immersed at Millbay marina while flushed with fresh water. This was the minimum salinity we used in an experiment to assess the mortality of fouling organisms attached to PVC panels when exposed to three different salinities (7, 20 and 33 psu (Control)). The 7 psu treatment was highly effective at killing most of the macrobenthos on the panels, whereas communities exposed to 20 and 33 psu were largely unaffected. There was some recolonization of bare substrata on the panels after one month, thus this treatment would be best carried out on sea chests before a vessel leaves port, if she is destined for another biogeographic region.

Freshwater exposure is an efficient way of controlling sublittoral marine fouling organisms as most suffer osmotic stress (Moreira et al., 2014; Quinton, 2014; Minto, 2014; Jude & Dunphy, 2016). Most organisms were killed by our two hour treatment with 7 psu water. For example, although *D. grossularia* had only 38% of mortality all the non-native *W. subatra* were all killed after one week. After one week many dead rotting organisms were seen, which then fell off the panels leaving bare space and revealing an understorey of organisms that were previously obscured, such as *Pomatoceros* sp. (Table 1).

Of the two commonest species found in this study, *D. grossularia* and *C. intestinalis*, the first is a small, robust tunicate, while the second is large, soft and highly contractile tunicate. Their bauplan possibly contributed to their differing vulnerability to the treatment. After one month, new *Clavelina lepadiformis* had colonized along with small erect bryozoans and *W. subatra* colonies (Table 1; Fig. 3). Thus flushing sea chests with seawater would be an effective treatment for removing biofouling but will be time-dependent, with new recruitment occurring within a month. For vessels which stay for long periods in berth we suggest low salinity flushing of sea chests is applied shortly before vessels depart for the next port of call.

**5 Conclusion**

Very high levels of mortality occurred in mature biofouling communities subjected to two hour treatment with 7 psu water, although some *Dendrodoa grossularia* were resilient. Low salinity treatments can be an efficient way of minimizing biofouling from ship sea-chests, and offer a promising tool to be incorporated in vessel operation. This would be an environmentally friendly biosecurity tool for minimizing and controlling ships sea-chest biofouling that is simple and would not cause undue delay or costs.

**Acknowledgments**





This study is part of a PhD research funded by the National Council for Scientific and Technological Development -
CNPq (grant award 200026/2015-1), an agency linked to the Ministry of Science and Technology, in charge of the
"Science without Borders Programme" with support from the Directorate-General for Nuclear and Technological
Development and the Directorate of Ports and Coasts of the Brazilian Navy.

**Disclosure statement**
The authors declare that they have no conflict of interest.

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





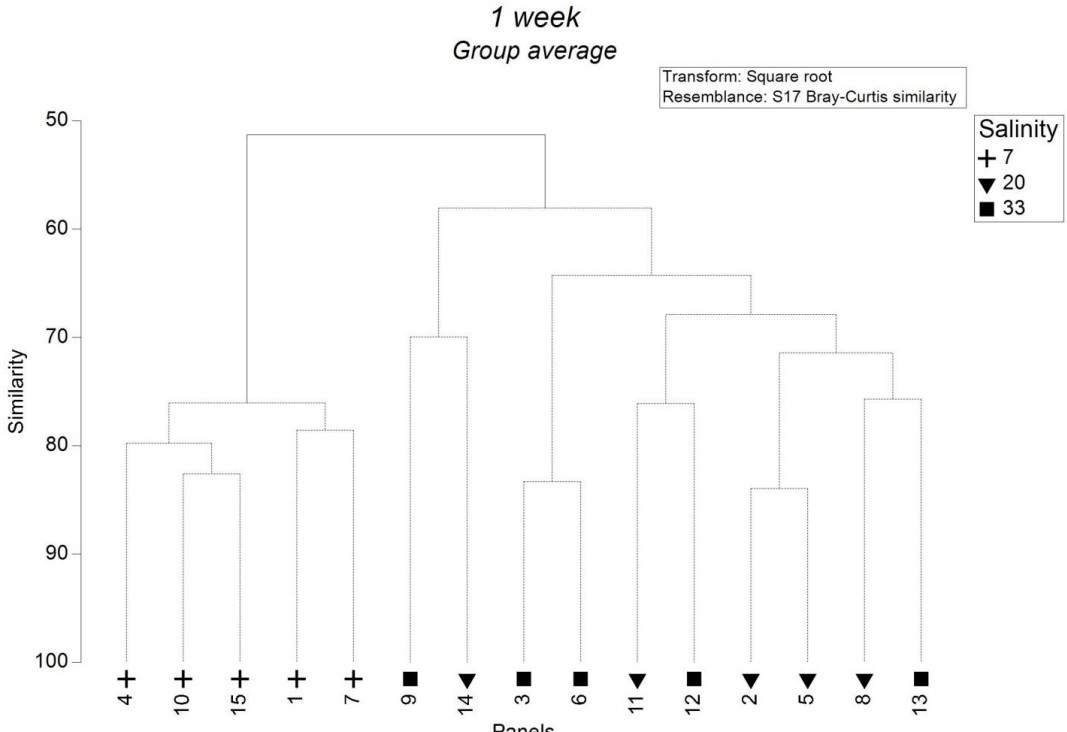

Figure. 1. Dendrogram showing significant separation between biofouling communities grown on settlement panels
treated with 7 psu and all the others treated with 20 psu and 33 psu (n=5 for each treatment).





Figure. 2. A) Settlement panel one week after exposure to a 33 psu treatment showing the high biomass and diverse
biofouling community that had developed over two years at 1.5 m depth in a marina off Plymouth, UK. B) Example
of a panel one week after exposure to a 20 psu treatment with many members of the biofouling community still
alive. C) Panel one week after a 7 psu treatment showing black sulphurous rotting tissues. D) Typical panel
appearance one month after exposure to 7 psu showing a much reduced fouling community.



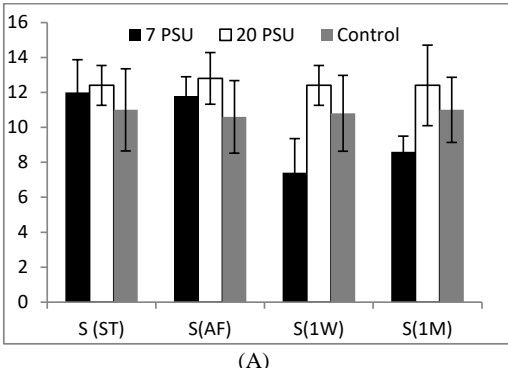
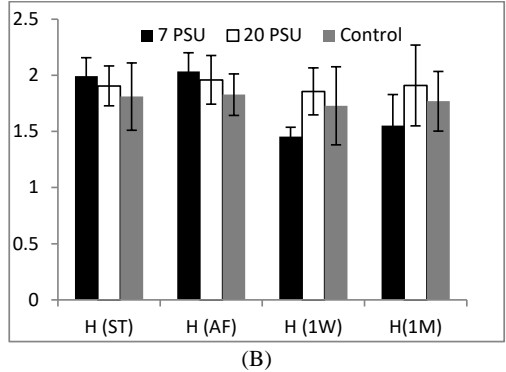

(A)  (B)

Figure 3. A) Average number of species and B) Shannon-Wiener diversity index (H') of two year old biofouling communities developed on PVC panels at 1.5 m depth in a marina off Plymouth, UK. Panels exposed to 7, 20 and 33 psu (Control) before treatment (ST), immediately after exposure (AF), one week after (1W) and after one month of exposure (1M). Error bars are ± SD, n=15.



Table 1: Average number of biofouling individuals per panel subjected to treatment with 7 psu, 20 psu and 33 psu (control) water, showing % change in abundance after one week and after one month.

| Taxa | Abundance data (average number of individuals/panel ± SD n=5) Pre treatment | | | % change after 1 week | | | % change after 1 month | | |
|---|---|---|---|---|---|---|---|---|---|
| | 7 psu | 20 psu | Control | 7 psu | 20 psu | Control | 7 psu | 20 psu | Control |
| Bare substratum | 8.2 ±3.0 | 4.2 ±5.3 | 5 ±3.3 | 404.9 | 142.9 | 40.0 | 385.4 | 109.5 | 48.0 |
| Biofilm | 27.2±11.7 | 28.6±14 | 23±10.4 | 21.3 | -2.8 | 3.5 | 83.8 | -15.4 | 13.9 |
| *Sycon ciliatum* | 0.4 | 3±1.7 | 4.6±6.2 | 0.0 | -33.3 | -17.4 | -100.0 | -80.0 | -78.3 |
| *Halichondria panicea* | 3.8±3.5 | 2±1.7 | 7.2±8 | 5.3 | 170.0 | -30.6 | 21.1 | 70.0 | -27.8 |
| *Corynactis viridis* | 1.6±4.2 | 0.0 | 0.0 | -100.0 | | | -75.0 | | |
| Sabellaridae | 2±1.5 | 1.2±0.6 | 0.2±0.7 | 0.0 | 0.0 | 100.0 | -100.0 | -16.7 | 0.0 |
| *Pomatoceros* sp. | 1±1.2 | 0.8 | 0.0 | 860.0 | 200.0 | | 1060.0 | 325.0 | |
| Terebellidae | 0.0 | 0.2 | 0.8±2.8 | | -100.0 | -50.0 | | -100.0 | -100.0 |
| *Watersipora subatra* | 1±1.2 | 0.8±1.4 | 0.4±0.6 | -100.0 | 75.0 | -100.0 | -20.0 | 225.0 | -100.0 |
| *Bugula neritina* | 7.8±4.6 | 8.6±9.7 | 8±11 | -100.0 | -67.4 | 57.5 | -46.2 | -51.2 | 42.5 |
| erect bryozoans | 12.6±8.6 | 10.6±8.3 | 12.6±10.7 | -100.0 | -32.1 | -23.8 | -58.7 | -30.2 | -33.3 |
| *Aplidium glabrum* | 1.6 | 0.0 | 0.0 | -100.0 | | | -25.0 | | |
| *Diplosoma listerianum* | 1±0.7 | 2.6±2.1 | 0.6±1 | -20.0 | -53.8 | 366.7 | -100.0 | 38.5 | 200.0 |
| *Botryllus schlosseri* | 0.8±1.4 | 0.0 | 0.0 | -100.0 | | | -100.0 | | |
| *Asterocarpa humilis* | 1.0 | 0.6 | 0.6±1 | -100.0 | 66.7 | -33.3 | -100.0 | 166.7 | -100.0 |
| *Styela clava* | 0.0 | 0.2 | 0.2±0.7 | | 0.0 | -100.0 | | -100.0 | -100.0 |
| *Corella eumyota* | 0.0 | 0.0 | 0.4±0.6 | | | -100.0 | | | -100.0 |
| *Clavelina lepadiformis* | 4.6±5.3 | 6.2±6.9 | 8.6±15.7 | -100.0 | -38.7 | -67.4 | 65.2 | 19.4 | -18.6 |
| *Ascidiella aspersa* | 7.2±8.7 | 7.8±7.3 | 3.8±5.1 | -88.9 | -46.2 | 10.5 | -100.0 | -23.1 | -63.2 |
| *Ascidia conchilega* | 0.0 | 0.0 | 0.2±0.7 | | | -100.0 | | | 0.0 |
| *Ascidia mentula* | 8.4 | 2.8±5.7 | 12.4±43.8 | -100.0 | -21.4 | 12.9 | -100.0 | -100.0 | 8.1 |
| *Ciona intestinalis* | 18.6±14.1 | 14.2±6.9 | 10.4±4.7 | -100.0 | -33.8 | -15.4 | -100.0 | -43.7 | 32.7 |
| *Dendrodoa grossularia* | 29±17.9 | 44.6±17.6 | 43.2±30.1 | -37.9 | 19.3 | 9.3 | -52.4 | 26.0 | 1.9 |