# Peer review of "Low salinity as a biosecurity tool for minimizing biofouling in"

_Ocean Science, 2018_

## Referee Comment (RC1) · Anonymous Referee #1 · 30 Mar 2018

Shipping is a major vector for transporting invasive species between ports, and hence an important factor in the introduction and establishment of such species. Biofouling and ballast water are the two vectors, and while the ballast water convention, which entered into force in 2017, deals with the ballast water and sediments in the tanks, there is no similar convention in force when it comes to biofouling.

Although hull is maybe the major surface area for biofouling, also algae and animals living and growing in the sea chests is a factor to take into account. This paper shows that exposing marine organisms to low salinity (7 psu) water in a (model) sea-chest will kill most of the organisms/taxa. The authors thus conclude that a low salinity treatment can be an efficient way of minimizing biofouling from ship sea-chests. To use freshwater as an in-water system to manage biofouling is not a new concept which has been

suggested earlier.

The paper addresses a relevant scientific question, but can be shortened and more to the point, and the cluster analysis does not add to the main conclusion: freshwater kills marine organisms. In the end, it all boils down to how can this (and other suggestions) be implemented in a way that works for a ship running a tight time schedule. I think the study will have a much heavier impact if the authors also include some practical test onboard ships, not to show that low saline water will kill the organisms, but how is it done? Or at least discusses it. Previous studies have both pointed out the possibility to use osmotic shock (freshwater), but also the practical problems (see e.g. Growcott et al 2016/2017, and references therein). I think this point must be expanded in the paper.

———————————————

---

## Author Comment (AC1) · 10 May 2018

Authors' answer to the anonymous reviewer's comments

Thank you very much for your constructive and technical comments.

Our aim is to show how effective low salinity is in killing biofouling from ships sea-chest. Although the use of freshwater has been proposed before, in our case we are proposing low salinity treatments for a short period of time is what makes it feasible for this to be used as a biosecurity tool to minimize biofouling from ships sea-chests.

Growcott et al. (2017) reviewed advantages and limitations of reactive systems to remove or treat biofouling in sea chests and internal pipework, and described a limitation for freshwater treatments is that they need a long exposure time and that biofouling

may remain attached to surfaces. This reference is certainly an important one and will be included in the manuscript. In our study, using a sea-chest model, we showed that just a couple of hours of exposure to low salinity waters killed macrobenthos that then became detached after one week.

As suggested by the reviewer we can remove the cluster analysis to simplify the manuscript.

As for ship board tests, logistics prevented us to from carrying out these experimental trials on-board and with a view to increasing replication for our tests we opted for a model sea chest. We also did not have budget for divers to carry out these tests on the merchant fleet.

---

## Referee Comment (RC2) · Anonymous Referee #2 · 13 Jun 2018

General comment

The paper tests the hypothesis that doses of low salinity water could be used to kill fouling species in ships sea-chests

The scope of the study is novel and data presented in the study of importance in the search for new antifouling methods for sea chests. From a biosecurity perspective the method is of interest as can help reduce risk of spreading invasive species. The new method proposed can also lead to a future decrease in use of biocidal (Cu) systems that presently are on the market to mitigate sea chest fouling (and I suggest that that aspect, of other sea chest mitigation alternatives, also is included, to balance the background with AF-paint historical overview)

However I think the authors clearly should state that it is somewhat a case study from a specific site (southern UK) and with the fouling community present in this region. This is a relevant fouling for the NE Atlantic but it should also be mentioned that fouling community composition can vary greatly and include for example higher percentage of hard fouling organisms like barnacles/mussels that have the possibility to close their calcareous shell when in unfavourable conditions (like low salinity).

Conclusions are ok but I think it would be good if limitations (of that the study was conducted in one single geographic area) and also with a fouling community grown under static condition and not in a sea chest onboard a ship in route, should be included.

Specific comments

At page 2 (Line 57) where said (e.g. bryozoan) suggest adding (e. g. bryozoan and algae) At page 2 (between Line 59-60) I suggest to include a sentence stating that the most commonly used AF paint systems used today are copper-based (as standing now it looks like the substances from marine organisms became the alternative to TBT, which is misleading. Here should also be explained the difference between biocides and boosterbicides before referring to the example study regarding soy-bean as one of the booster biocides (there are numerous booster biocides currently in use in commercial paints with role to be effective against algal fouling)

Page 3 (Line 88) "low saline environment can kill fouling species" is described a bit short. I suggest a sentence is added with a more developed resoning. Killed by what mechanism (osmosis) and that it is the rapid change in salinity (not necessarily the low salinity?) that are stressful but the change from the environment where the fouling was developed.

Page 4 (Line 123) at what temperature was the organisms stored (compared to the marina water temp)?

Page 4 (Line 143) I do not get the "onwards" in this sentence, please clearify

---

## Author Comment (AC2) · 19 Jun 2018

Authors' answer to Referee 2 comments:

Thank you very much for your comments and suggestions which certainly improve substantially our manuscript. All of them are addressed directly in the text of the manuscript (in red) as follows:

[revised manuscript text omitted]

---

## Editor Comment (EC1) · D. Turner (Editor) · 20 Jun 2018

The practical salinity scale is dimensionless, i.e. there is no unit "psu". Salinity should therefore be given as a number without unit.

---

## Author Comment (AC3) · 20 Jun 2018

Thank you for your comment. When a new text of the manuscript is required with all the suggestions / comments received during the discussion period, we'll delete 'psu' from our salinity values, keeping only the algarisms.